# Genome-Wide Identification of *Ginkgo biloba* SPL Gene Family and Expression Analysis in Flavonoid Biosynthesis and Water Stress

**DOI:** 10.3390/ijms26104932

**Published:** 2025-05-21

**Authors:** Meiling Ming, Mulin Yi, Chunyue Qin, Luyao Yan, Yuhan Sun, Juan Zhang, Fuliang Cao, Fangfang Fu

**Affiliations:** State Key Laboratory of Tree Genetics and Breeding, Co-Innovation Center for Sustainable Forestry in Southern China, Nanjing Forestry University, Nanjing 210037, China; mingmeiling@njfu.edu.cn (M.M.); yimumu@njfu.edu.cn (M.Y.); qinchunyue@njfu.edu.cn (C.Q.); yanluyao@njfu.edu.cn (L.Y.); jianjiao@njfu.edu.cn (Y.S.); juanzhang@njfu.edu.cn (J.Z.)

**Keywords:** *Ginkgo biloba*, SPL, gene family, flavonoid biosynthesis, water stress

## Abstract

SQUAMOSA promoter-binding protein-like (SPL) transcription factors specific to plants are vital for regulating growth, development, secondary metabolite biosynthesis, and responses to both biotic and abiotic stresses. Despite their importance, no systematic investigations or identifications of the SPL gene family in *Ginkgo biloba* have been conducted. In this study, we identified 13 SPL genes within the *Ginkgo biloba* reference genome, spanning seven chromosomes, and categorized these genes into six groups based on their phylogenetic relationships with *Arabidopsis thaliana* SPL gene families. Our analysis of gene structure, conserved domains, motifs, and miR156 target predictions indicates that GbSPLs are highly conserved across evolutionary timelines. Furthermore, synteny analysis highlighted that dispersed duplication events have expanded the SPL gene family in *Ginkgo biloba*. Examination of the cis-regulatory elements revealed that many *GbSPL* genes possess motifs associated with light, hormones, and stress, implying their involvement in flavonoid biosynthesis and adaptation to environmental conditions. RNA-Seq and qRT-PCR expression profiles of *GbSPL* genes across various tissues and low- and high-flavonoid leaves and during both short-term and long-term water stress illustrated their roles in flavonoid biosynthesis and responses to water stress. Subcellular localization experiments showed that GbSPL2 and GbSPL11 proteins are situated within the nucleus. Our research offers the first systematic characterization of the SPL gene family in *Ginkgo biloba*, establishing a valuable foundation for understanding their evolutionary background and functional roles in flavonoid biosynthesis and water stress response.

## 1. Introduction

Transcription factors (TFs) are a group of proteins that contain structural regions capable of binding to specific DNA sequences located in gene promoter regions. They act as molecular switches that either enhance or inhibit the transcription of various genes [1]. Previous studies have characterized multiple transcription factor families, including R2R3-myeloblastoma (MYB), basic helix-loop-helix (bHLH3), WD40, APETALA2/ethylene-responsive factor (AP2/ERF), basic leucine zipper (bZIP), MADS-box, WRKY, NAC, homeodomain-leucine zipper (HD-Zip), and SQUAMOSA promoter-binding protein-like (SPL) transcription factors [2]. SPL proteins are plant-exclusive transcription factors characterized by their 76-residue SBP domain. This evolutionarily conserved domain incorporates dual zinc finger motifs (C3H and C2HC types) and a bipartite nuclear localization signal (NLS). Notably, the C-terminal C2HC zinc finger spatially overlaps with the NLS, forming a structural module that enables nuclear import and subsequent gene regulation [3].

The initial identification of SPL genes (*AmSBP1* and *AmSBP2*) took place in the floral meristems of snapdragon (*Antirrhinum majus*) in 1996 [4]. Genome-wide studies have subsequently annotated SPL homologs across plant species: 17 members in *Arabidopsis thaliana* [5], 19 members in rice (*Oryza sativa*) [5], 15 members in tomato (*Solanum lycopersicum*) [6], 28 members in poplar (*Populus trichocarpa*), 27 members in the apple (*Malus domestica Borkh.*) [7], 24 members in Tartary Buckwheat (*Fagopyrum tataricum*) [8], 20 members in *Betula platyphylla Suk.* [9], 19 members in pear (*Pyrus pyrifolia*) [10], 23 members in *Medicago truncatula* [11], 17 members in sugarcane (*Saccharum spontaneum*) [12], 23 members in quinoa (*Chenopodium quinoa*) [13], 49 members in *Nicotiana tabacum*, 43 members in *Nicotiana benthamiana* [14], 25 members in sunflower (*Helianthus annuus*) [15], 12 members in sweet cherry (*Prunus avium* L.) [16], 17 members in litchi (*Litchi chinensis Sonn.*) [17], and so on. The classification of SPL gene subfamilies is largely consistent across different species and can generally be divided into six to nine subfamilies.

The SPL gene family typically operates through independent mechanisms or pathways that miR156 targets. The evolutionarily conserved miR156 family post-transcriptionally regulates multiple SPL genes by specifically cleaving their mRNA transcripts, which is a hallmark regulatory mechanism in plant development, secondary metabolism, and stress response [18]. The 20-nucleotide miR156 was originally characterized in *Arabidopsis thaliana*, and subsequent functional studies elucidated its regulatory partnership with SPL transcription factors (miR156-SPL module) in modulating pigmentation patterns across plant tissues [19,20]. For instance, SPL transcripts act as direct targets of miR156 in European aspen (*Populus tremula* L.), with this regulatory interaction influencing the outputs of the phenylpropanoid pathway, including anthocyanins, flavonoids, and flavonols [21]. In addition to their role in plant secondary metabolism, the miR156/SPL module has also been shown to play a crucial part in plant growth and development. Emerging evidence identifies the miR156-SPL axis as a key regulator of root system architecture. Molecular phenotyping indicates that the depletion of miR156 inhibits both lateral root initiation and adventitious root formation through SPL-mediated transcriptional reprogramming [22]. Cutting-edge plant synthetic biology strategies now use miR156 network rewiring to create stress-resilient crops, combining target gene editing with advanced transformation platforms for abiotic adaptation. The overexpression of miR156 enhanced stress tolerance in *Arabidopsis thaliana* and rice, uncovering an evolutionarily conserved miR156-SPL-DFR regulatory axis that aligns developmental plasticity with abiotic stress tolerance, forming a molecular basis for environmental adaptation in plants [23]. In addition to the miR156-SPL module, six AtSPLs in Arabidopsis—*AtSPL1*, *AtSPL7*, *AtSPL8*, *AtSPL12*, *AtSPL14*, and *AtSPL16*—are unaffected by miR156. Importantly, *AtSPL7* binds directly to the Cu-response element (CuRE) featuring the core sequence GTAC, which is essential for regulating copper homeostasis in the plant [24]. Additionally, both targeted and non-targeted SBP-box transcription factors work together to promote male fertility in Arabidopsis [25]. Overexpressing LcSPL2 in *Arabidopsis thaliana* advanced flowering and reduced rosette leaf number. Additionally, elevated LcSPL2 expression in *A. thaliana* increased the transcript levels of four flower-development-related genes, suggesting the role of LcSPL2 in promoting flower development [26]. OsSPL10, a transcription factor implicated in enhancing drought resistance in rice, plays a crucial role in this process by regulating the production of reactive oxygen species (ROS) and the movement of stomata. Analysis of haplotype and allele frequencies revealed that the OsSPL10Hap1 allele is prevalent in upland and improved lowland rice varieties, while the OsSPL10Hap2 allele is primarily found in lowland and landrace rice cultivars [27]. Furthermore, MeSPL9 impacts drought resistance by regulating levels of protective metabolites and JA signaling [28], while wheat plants overexpressing TaSPL6-A showed increased susceptibility to drought stress [29].

While SPL transcription factors are recognized regulators of various developmental pathways in angiosperms, their functional characterization in *Ginkgo biloba* remains significantly understudied. As a paleoendemic species often referred to as a “living fossil”, *Ginkgo biloba* is widely cultivated worldwide for its dual benefits: medicinal properties and distinctive fan-shaped leaves. The medicinal properties of *Ginkgo biloba* primarily rely on secondary metabolites such as flavonoids and terpenoids. Ginkgo leaf extracts are utilized in the healthcare and food sectors because *G. biloba* flavonoids and their glycosides exhibit notable effects against bacteria, viruses, inflammation, and cancer while also offering neuroprotection [30]. Ginkgo trees amass antimicrobial and antioxidant secondary metabolites in the xylem, such as flavonoid glycosides and terpenes, which contribute to their exceptional longevity [31]. Flavonoid compounds can enhance *G. biloba*’s resistance to soil salt stress. Exposure to UV-B radiation also boosts the overall content of flavonoid glycosides [32]. *Ginkgo biloba* also demonstrates remarkable resilience to biotic challenges and abiotic stressors, with exceptional drought tolerance rooted in its evolutionary background. Thirteen putative GbSPL genes were identified in this study. A comprehensive analysis was conducted on gene properties, phylogenetics, structure, collinearity, cis-acting elements, and gene expression patterns. Genes related to flavonoid biosynthesis and water stress responses were prioritized for detailed examination. The findings of this study establish a framework for unraveling the transcriptional regulation of flavonoid metabolism and water stress in *Ginkgo biloba*. Comparative analysis of stress adaptation genes across ginkgo accessions may uncover conserved genetic modules suitable for pyramiding breeding, thereby synergistically enhancing the yields of medicinal compounds and drought-resistant traits.

## 2. Results

### 2.1. Whole-Genome Identification of the GbSPL Genes in Ginkgo biloba

The SPL gene family in *Ginkgo biloba* was systematically characterized using two independent approaches: the BLAST method and the HMMsearch method. Both methods identified 13 candidate genes each. The 13 overlapping members between the BLAST and HMMsearch methods were considered GbSPL genes in *Ginkgo biloba*. Subsequently, we removed the member with incomplete SBP domains based on the NCBI CDD website. Finally, a total of 13 GbSPL genes with the conserved SBP domain were identified in *Ginkgo biloba* and renamed according to their chromosomal positions (Table 1). Substantial variation was observed in protein properties: amino acid lengths ranged from 170 aa (GbSPL4) to 1314 aa (GbSPL10), with corresponding molecular weights varying from 18,774.35 Da to 145,251.18 Da (Table 1). Protein isoelectric points (pIs) showed values between 5.57 (GbSPL13) and 9.27 (GbSPL9). Additionally, their grand average of hydropathicity indices ranged from −0.776 to −0.414, and the instability index and aliphatic index were also analyzed. The instability index ranged from 37.22 (GbSPL8) to 65.54 (GbSPL6); only GbSPL8 was a stable protein. Predictions for subcellular localization indicated nuclear-specific localization for all SPL proteins except for the chloroplast localization of GbSPL10. miR156 target predictions revealed that only two GbSPLs (GbSPL1 and GbSPL9) possess GbmiR156 binding sites (Table 1).

The chromosomal mapping of 13 *GbSPL* genes in *Ginkgo biloba* revealed their uneven genomic arrangement, visualized through TBtools-II software for spatial distribution analysis. Overall, these 13 *GbSPL* genes were mapped across seven chromosomes: chr1, chr5, chr6, chr8, chr9, chr10, and chr11 (Figure 1). Both chr1 and chr5 contain three *GbSPL* genes each, while chr6, chr8, and chr10 each contain one *GbSPL* gene.

### 2.2. Phylogenetic Analysis of GbSPL Genes

To investigate the evolutionary relationship of the identified *GbSPL* genes in *Ginkgo biloba*, a Maximum Likelihood (ML) phylogenetic tree was constructed using the protein sequences of 13 GbSPL genes from *Ginkgo biloba*, 16 AtSPL genes from *Arabidopsis thaliana*, and 19 OsSPL genes from *Oryza sativa* (Figure 2). The results revealed that the SPL gene families were classified into eight groups (Group I to Group VIII), consistent with a previous classification of SPL family distribution [8,17]. The GbSPL gene family was distributed across six groups (Group I, II, III, IV, V, VIII): one member in Group I, two members in Group II, five members in Group III, one member in Group IV, two members in Group V, and two members in Group VIII (Figure 2). This result suggested that the functions of SPL genes might have diverged during the evolution of *Arabidopsis thaliana* (dicotyledonous), *Oryza sativa* (monocotyledonous), and *Ginkgo biloba* (gymnosperm).

### 2.3. Motif, Conserved Domain, and Gene Structure Analysis of GbSPLs

To further provide insight into the GbSPLs, we investigate the motif, conserved domain, and gene structure analysis of GbSPL gene family members. The clustering was constructed based on the results of the phylogenetic analysis of SPL in *Ginkgo biloba* (Figure 3A). We utilized the MEME website tool to analyze the differences among the 10 conserved motifs (Motif 1–Motif 10) of the GbSPL proteins (Figure 3B and Appendix A). Only motif 1 was present in all 13 *GbSPL* genes, whereas motif 2 was identified in 10 *GbSPL* gene members. Motif 1 and motif 2 may constitute parts of the conserved domain of SBP (Appendix A). The conserved motifs indicated that the same subgroup of GbSPL exhibited similar types, while different subgroups of GbSPL displayed varying motif types. For example, GbSPL3 and GbSPL8 in group V both contain motifs 1, 4, and 7. However, their motif composition shows notable differences compared to the members of group II, GbSPL7 and GbSPL10. Notably, despite being clustered together, the five members within group III (GbSPL2, 5, 6, 11, 12) exhibit discernible variations in their motif compositions. These variations within the same subgroup suggest that genes in the same subgroup might exhibit functional diversity.

We also identified the conserved domains of the GbSPL gene family members. The SBP domain was present in all 13 GbSPL family members. Structural conservation was noted in Group II members (GbSPL7 and GbSPL10), each containing both SBP and Ank domains (Figure 3C). Given the established role of the Ank domain in orchestrating protein interactomes through molecular recognition interfaces [25], GbSPL7 and GbSPL10 likely function as signaling nodes in cellular communication networks.

Gene structure is an important feature that can influence gene expression and function. In this study, we measured the exon–intron compositions of the 13 *GbSPL* genes (Figure 3D). The number of exons (CDSs) varied from 2 (GbSPL3 and GbSPL4) to 14 (GbSPL10), while the number of introns ranged from 1 to 13. This difference in gene structure indicates the functional diversity of *GbSPL* genes. Although the core coding sequences are well-defined, a comprehensive annotation of regulatory elements, including the 5′-UTR and 3′-UTR, remains a critical gap in the current *Ginkgo biloba* genomic resources [33].

### 2.4. Multiple Sequence Alignment Analysis and MicroRNA Target Prediction

Comparative SBP domain mapping identified three signature elements across the 13 GbSPLs: two zinc-binding clusters (Zn-1[C3H] and Zn-2[C2HC]) and an NLS (Figure 4A). Sequence conservation analysis showed that the SBP domains were highly conserved at the CQQC, SCR, and RRR sequences, with approximately 76 amino acid residues (Figure 4A). Considering the importance of the miR156/SPL module [20], the miR156 target sites of the 13 GbSPL genes were also predicted using the psRNATarget online tool. Surprisingly, only 2 out of 13 SPL genes contained miR156 binding sites. Each of these two genes harbored dual target loci: one recognized by miR156abc and the other by miR156f (Figure 4B).

### 2.5. The Expansion and Collinearity Analysis of the GbSPL Gene Family

MCScanX software (v1.0.0) was utilized to identify the expansion and evolution history of the GbSPL gene family. A total of 7285 (26.2%) genes in the whole genome were generated through singleton duplication; 12,615 (45.3%) genes were generated by dispersed duplication; 1688 (6.1%) genes were generated through proximal duplication; 4905 (17.6%) genes were produced by tandem duplication; and 1343 (4.8%) genes were produced by whole-genome duplication (WGD) or segmental duplication (Figure 5A). Among the 13 GbSPL gene members, seven GbSPL genes resulted from dispersed duplication, four GbSPL genes originated from tandem duplication, and two GbSPL genes stemmed from WGD or segmental duplication. At the same time, none were produced by singleton or proximal duplication (Figure 5A,B). Two syntenic gene pairs (GbSPL3-GbSPL8, GbSPL2-GbSPL5) were identified within the GbSPL gene family (Figure 5C). These results indicate that dispersed duplication events were the primary force behind expanding the GbSPL gene family.

To elucidate the evolutionary dynamics of SPL gene duplication, cross-species synteny analyses were conducted comparing *Ginkgo biloba* with both a herbaceous model (*Arabidopsis thaliana*) and a woody relative, Poplar (*Populus alba* × *Populus tremula* var. *glandulosa* clone ‘84K’), utilizing MCScanX for whole-genome collinearity detection. Overall, ginkgo shows less genome-wide covariance with *Arabidopsis thaliana* and possesses two covariant SPL genes among them; it has no covariant SPL gene with the woody plant Poplar, although there are more homologous gene pairs with Poplar than with Arabidopsis in the whole genome (Figure 5D). Despite their more recent divergence as woody species, *Ginkgo biloba* and Poplar exhibit ancient SPL gene architectures, suggesting prolonged conservation of this regulatory family. Cross-species comparisons of GbSPL homologs could clarify functional divergence patterns while informing ancestral gene repertoire reconstructions in angiosperms and gymnosperms.

### 2.6. Cis-Acting Element Analysis in the Promoter of GbSPL Genes

Cis-regulatory elements within promoter regions mediate transcriptional modulation, thereby governing phenotypic expression through precise spatiotemporal control of gene activity. An in silico promoter analysis was conducted on GbSPL genes using the PlantCARE database, systematically screening for cis-acting elements within 2000 bp upstream of transcription start sites. A diverse distribution of cis-elements was observed in each GbSPL, including MYB binding sites, light, auxin, gibberellin, MeJA, abscisic acid, defense and stress responses, low-temperature elements, and others (Figure 6A), suggesting that GbSPLs may be involved in various biological processes. The cis-elements were categorized into four groups: plant growth and development-responsive elements (CAT-box, O2-site, MSA-like, and RY element), hormone-responsive elements (AuxRR-core, ABRE, TCA-elements, TATC-box, GARE-motif, CGTCA motif, TGACG motif, P-box, and TGA-element), light-responsive elements (TCT-motif, G-box, 3-AF1 site, GT1 motif, AAAC motif, ATCT motif, G-box, Sp1, GA motif, CAG motif, I-box, GATA motif, Gap box, ACE, LAMP element, MRE, ATC motif, TCCC motif, AE-box, chs CMTA, and AT1 motif), and stress-responsive elements (ARE, LTR, MBS, GC motif, and TC-rich repeats) (Figure 6B). The largest category of cis-elements was light-responsive, while hormone-responsive and stress-responsive elements were also prevalent in the promoters of GbSPL genes. Plant hormones, such as ABA and gibberellin, have been recognized as crucial components in various stress signal responses, and these cis-acting elements on the GbSPL promoters contribute to their functional diversity under stress conditions. The cis-acting analysis indicated that GbSPL genes play significant roles in both light and stress responsiveness.

### 2.7. Expression Profiles of GbSPL Genes in Various Tissues, Low- and High-Flavonoid Leaves, and Water Stress in Ginkgo biloba

First, we investigated the tissue-specific expression patterns using RNA-Seq data from the reported study [34]. RNA-Seq-based expression profiles showed that *GbSPL* genes were differentially expressed in eight tissues, including mature fruit (MF), immature fruit (IF), root (R), stem (S), immature leaf (IL), mature leaf (ML), microstrobilus (M), and ovulate strobilus (OS) (Figure 7A). All 13 *GbSPL* genes were highly expressed in the stem, and 8 *GbSPL* genes (*GbSPL1*, *5*, *6*,*7*, *9*, *10*, *11*, *13*) exhibited high expression in the root (Figure 7A). The expression patterns of *GbSPL* genes in mature and immature fruits, as well as in mature and immature leaves, were highly consistent, indicating that the expression of the SPL gene is tissue-specific but not maturation-stage-specific. Additionally, most GbSPL genes were highly expressed in the ovulate strobilus but not in the microstrobilus, suggesting their role in the reproductive phase change.

Next, we explored the expression profiles of GbSPLs in four groups contrasting low and high flavonoid content [32,35,36,37]. Overall, the differentially expressed *GbSPL* genes varied across each group. For instance, *GbSPL1*, *GbSPL2*, *GbSPL11*, and *GbSPL12* were downregulated in the high-flavonoid leaves (UV-B treatment) of the first low- and high-flavonoid group (control and UV-B treatment) but not in the other groups (Figure 7B). Additionally, *GbSPL10* was specifically downregulated in the high-flavonoid leaves (yellow mutant) of the third low- and high-flavonoid group (green and yellow-mutant leaves), while *GbSPL3* was uniquely downregulated in the high-flavonoid leaves (‘ZY1’) of the fourth low- and high-flavonoid group (‘Anlu1’ and ‘ZY1’ leaves). These findings suggested that the biosynthetic and regulatory networks of flavonoids may differ based on the types of low and high flavonoids, possibly requiring the involvement of multiple SPL gene types.

Finally, we detected the expression patterns of *GbSPL* genes under both short-term and long-term water stress. During short-term water stress and rehydration (0, 3, 6, 12, 24, Re12 h) [38], the *GbSPL* genes were classified into two clusters: one induced by water stress (*GbSPL4*, *8*, *12*) and the other repressed by water stress (*GbSPL1*, *2*, *5*, *7*, *10*, *11*, *13*) (Figure 7C). In the long-term drought treatment (0, 15, 22 d, and re-watering 3 d) [39], the *GbSPL* genes were similarly divided into two clusters: one induced by drought stress (*GbSPL3*, *7*, *10*, *11*, *13*) and the other slightly repressed by drought stress (*GbSPL1*, *2*, *4*, *5*, *8*) (Figure 7D). Significant differences exist between short-term and long-term water stress in the upregulated and downregulated genes. This is likely due to distinct regulatory networks that govern responses to short-term and long-term water stress in *Ginkgo biloba*. For instance, *GbSPL11* was downregulated during short-term water shock but upregulated under long-term drought stress, while *GbSPL2* was downregulated in both short-term water shock and long-term drought stress (Figure 7C,D).

### 2.8. Quantitative Real-Time PCR (qRT-PCR) Analysis and Subcellular Localization Assay of the GbSPL Genes

To validate the results of RNA-Seq, four genes (*GbSPL2*, *10*, *11*, and *13*) that showed differential expression in short-term or long-term water stress responses were selected for qRT-PCR analysis. The qRT-PCR results were similar to the RNA-Seq findings, suggesting that *GbSPL2*, *10*, *11*, and *13* may function as negative regulators of short-term water shock in *Ginkgo biloba* (Figure 8A). However, the expression levels of *GbSPL2*, *10*, *11*, and *13* were upregulated during long-term drought stress compared to the control (Figure 8B), which is also consistent with the RNA-Seq results.

To investigate the location of GbSPL proteins, we examined the subcellular localization of GbSPL2 and GbSPL11 through transient expression in tobacco (*Nicotiana benthamiana*) leaves. Microscopic visualization revealed that the signal for the empty vector control, 35S-GFP, was distributed in both the nucleus and the membrane. We observed that the green fluorescent protein (GFP) signals of GbSPL2 and GbSPL11 overlapped with the nuclear marker mCherry, indicating their localization in the nucleus (Figure 8C), which is consistent with the prediction results from the WoLF PSORT online tools (Table 1). These findings indicate that GbSPL2 and GbSPL11 function as nuclear proteins, possibly serving as transcription factors.

## 3. Discussion

### 3.1. Characteristics and Evolution of SPL Genes in Ginkgo biloba

As key plant-specific transcriptional regulators, SPL genes orchestrate multifaceted developmental programs encompassing secondary metabolite biosynthesis, morphogenesis (leaf and flower architecture), phase transition (juvenility–adulthood), reproductive timing control, and fruit morphogenesis through evolutionarily conserved molecular mechanisms [18,40]. The evolutionary trajectory of SPL genes dates back to their initial characterization in *Antirrhinum majus*, with subsequent phylogenetic analyses revealing conserved orthologs across Viridiplantae lineages—from basal streptophytes (Chlorophyta algae, Bryopsida mosses) to derived angiosperms (*Arabidopsis thaliana*, crop species, and woody plants) [17,41]. The functional conservation and divergence of SPL homologs in gymnosperms—evolutionarily significant non-flowering seeds plants—remain underexplored compared to angiosperm systems. *Ginkgo biloba* is an important gymnosperm whose SPL gene has not been reported. In this experiment, 13 *GbSPL* genes from *Ginkgo biloba* were screened using both a genome-level database constructed from the Pfam database (PF03110) and the BLAST protein database (Table 1). The number of GbSPL gene family members is comparable to that of *Arabidopsis thaliana* (16 members), tomato (15 members) [6], and Sweet Cherry (12 members) [16]. Moreover, the chromosomal mapping of the 13 *GbSPL* genes in *Ginkgo biloba* revealed their uneven distribution across seven chromosomes (Figure 1), which is also similar to litchi [17]. The phylogenetic analysis showed that the GbSPL gene family was only distributed across six groups, whereas the AtSPL gene family was classified into eight groups (Figure 2), suggesting that the function of SPL genes may have diverged during the evolution of *Arabidopsis thaliana* (dicotyledonous) and *Ginkgo biloba* (gymnosperm).

The phylogenomic integration of sequence homology, motif analysis, gene structure, and molecular evolution patterns demonstrates conserved taxonomic clustering among *Ginkgo biloba* species. GbSPL genes within phylogenetic clades maintain conserved structural blueprints (Figure 3), implying evolutionary selection pressures linking gene structure plasticity to functional diversification [42]. GbSPL5 and GbSPL6 exhibit the same three motifs (Figure 3B), possibly due to internal tandem duplications on the same chromosome. Each GbSPL gene contains an SBP domain, whereas only GbSPL7 and GbSPL10 include an Ank domain (Figure 3C). In addition, target prediction identified 2 out of 13 GbSPL genes (GbSPL1 and GbSPL9) with miR156 target sites that may function within the miR156-SPL module (Figure 4B). However, neither GbSPL1 nor GbSPL9 possesses the Ank domain (Figure 3C). These results suggest functional divergence within the GbSPL family: while GbSPL1 and GbSPL9 likely coordinate with miR156, forming miR156-SPL modules to orchestrate growth-related processes, the majority of GbSPLs appear to operate through miR156-independent regulatory circuits. These findings differ from those of litchi [17] and *Medicago truncatula* [11], as the majority of SPL genes in litchi and *Medicago truncatula* contain the target sites for miR156.

We also investigated the duplication events of the GbSPL gene in the whole genome of *Ginkgo biloba* and found that the dispersed duplication is the lead duplication event, followed by tandem duplication and WGD or segmental duplication; however, no singleton or proximal duplication events were found (Figure 5A,B). This suggests that transposable element-mediated dispersed replication plays a major role in the expansion of the GbSPL gene family. Tandem and WGD or segmental duplications lead the gene family amplification in plants [43], which are also popular in expanding the GbSPL gene family. In addition, collinearity analysis with *Arabidopsis thaliana* and Poplar (*Populus alba* × *Populus tremula* var. *glandulosa* clone ‘84K’) showed that the GbSPL gene had more homologous genes with *Arabidopsis thaliana* than with Poplar (Figure 5D), suggesting prolonged conservation of the SPL regulatory family.

### 3.2. Expression Patterns and Potential Functions of the GbSPL Genes

Gene expression profiles generally exhibit a strong correlation with biological functions. Cis-elements in the promoter regions of SPL genes are closely linked to expression profiles and function, as they are typically involved in regulating gene transcription [44]. Research shows that many cis-elements in the promoter regions of SPL genes are associated with plant growth and development, light, hormone, and stress responses (Figure 6), similar to those in other plants, such as *Betula luminifera* [45] and litchi [17]. Developmental time-course analyses in Arabidopsis revealed the stage-specific upregulation of *AtSPL15*, peaking during floral initiation phases with marked accumulation in inflorescence meristems [46]. Functional studies further demonstrated that *AtSPL15* accelerates the vegetative phase change while suppressing rosette leaf expansion. To predict the function of the GbSPL genes in plant growth and development, RNA-Seq-based expression patterns among various tissues and organs were first examined (Figure 7A). Most *GbSPL* genes exhibited high expression in the ovulate strobilus but not in the microstrobilus, suggesting their role in the reproductive phase change. Eight *GbSPL* genes (*GbSPL1*, *5*, *6*, *7*, *9*, *10*, *11*, *13*) showed elevated expression in the root, indicating that they may play a novel role in root development.

Flavonoids are the key active compounds in *Ginkgo biloba*, and their biosynthesis is influenced by light and hormones [47]. Considering that many cis-elements in the promoter regions of SPL genes are associated with light and hormones, we speculated that the GbSPL genes are related to flavonoid biosynthesis. Analysis of targeted metabolite and flavonoid biosynthetic gene expression has demonstrated that *OsSPL17* regulates the expression of flavonoid biosynthesis genes, specifically CHI [48]. To predict the function of the *GbSPL* genes in flavonoid biosynthesis, expression patterns among the low- and high-flavonoid content were investigated in four groups (Figure 7B). Transcriptional profiling revealed many *GbSPL* genes that were differentially expressed in the four low- and high-flavonoid content groups (e.g., *GbSPL1*, *GbSPL2*, *GbSPL10*, *GbSPL11*, and *GbSPL12*). However, non-overlapping sets of differentially expressed *GbSPL* genes across these groups indicated group-specific regulatory dynamics.

Many important and diverse regulatory functions of SPL genes in plant growth and development have been well-understood [18]. However, the role of SPL genes in response to abiotic stress remains insufficiently explored. Given our interest in water stress, we also examined the gene expression of all GbSPL members under both short-term water shock conditions and long-term drought stress conditions (Figure 7C,D). The results revealed the reduced expression of some *GbSPL* genes (e.g., *GbSPL2*) in both short-term water shock and long-term drought stress, while the expression patterns of other *GbSPL* genes (*GbSPL4*, *8*, *11*, *12*) were completely opposite in short-term and long-term drought conditions. This indicates that the *GbSPL4*, *8*, and *12* genes may respond quickly and help ginkgo resist damage from adverse conditions in the short term, but they appear to lack function during prolonged drought conditions. The genes that respond to short-term and long-term water stress are not exactly the same, which is similar to beet (*Beta vulgaris* L.) [49] and foxtail millet (*Setaria italica*) [50].

## 4. Materials and Methods

### 4.1. Whole-Genome Identification of SPL Genes in Ginkgo biloba

The high-quality genomic data of *Ginkgo biloba*, as referenced in source [33], were procured from the Genome Sequence Archive database (https://ngdc.cncb.ac.cn/gwh/Assembly/18742/show, accessed on 1 April 2025). Initially, the amino acid sequences corresponding to the SPL gene family members in *Arabidopsis thaliana* were acquired from the Arabidopsis Information Resource (TAIR) database (https://v2.arabidopsis.org/, accessed on 1 April 2025). The complete protein sequences of *Ginkgo biloba* were utilized to establish a BLAST protein database using the makeblastdb software (V 2.10.1). By employing the Arabidopsis SPL genes as queries in a BLASTp (V 2.10.1) search against the genome-wide protein database of *Ginkgo biloba*, candidate SPL members were identified under stringent criteria (E-value < 1 × 10^−5^, Identity > 30%). Subsequently, the Hidden Markov models (HMMs) related to the SBP domain (PF03110) were retrieved from the Pfam database (http://pfam.xfam.org/, accessed on 1 April 2025). The HMMsearch software (V 3.3.2) was used to examine the complete protein sequences of *Ginkgo biloba* based on the Pfam-A models file containing the SBP domain, which also yielded a list of candidate SPL members of *Ginkgo biloba*. Thirdly, the overlapping candidate SPL genes were delineated by intersecting the results derived from both BLAST and HMM searches. Lastly, the NCBI CDD website tool (https://www.ncbi.nlm.nih.gov/Structure/bwrpsb/bwrpsb.cgi, accessed on 1 April 2025) was utilized to verify the completeness of the SBP domains within the SPL genes. The GbSPL genes with the conserved SBP domain were identified in *Ginkgo biloba* and renamed according to their chromosomal positions.

### 4.2. Physicochemical Properties, MicroRNA Target Prediction, and Chromosomal Location Analysis of SPL Genes

The protein-coding sequence length, molecular weight (MW), and isoelectric point (pI) were calculated using ExPASy ProtParam online tools (https://web.expasy.org/protparam/, accessed on 1 April 2025), and subcellular localization predictions of GbSPL genes were conducted with WoLF PSORT (https://wolfpsort.hgc.jp/, accessed on 1 April 2025). Using default settings, the miR156 target sequences were predicted with the online tool psRNATarget (https://www.zhaolab.org/psRNATarget/home, accessed on 1 April 2025). The chromosomal localization of the GbSPLs was obtained from *Ginkgo biloba* genome annotation information and visualized by TBtools-II (V2.210) [51].

### 4.3. Phylogenetic Analysis

Muscle software (V 3.8.1551) was used to perform multiple sequence alignments of SPL genes in *Arabidopsis thaliana* and *Ginkgo biloba*. We constructed the phylogenetic tree using IQ-Tree with the Maximum Likelihood (ML) method and 1000 bootstrap iterations. The final tree topology was visually rendered and annotated with the iTOL web server (https://itol.embl.de/, accessed on 1 April 2025).

### 4.4. Motif, Conserved Domain, Gene Structure, and Cis-Elements Analysis

MEME (https://meme-suite.org/meme/tools/meme, accessed on 1 April 2025) was utilized to search for the 10 motifs (6–50 width; any number of repetitions (anr)) in the *Ginkgo biloba* SPL family proteins. The NCBI Batch CD-Search Tool (https://www.ncbi.nlm.nih.gov/Structure/bwrpsb/bwrpsb.cgi, accessed on 1 April 2025) was employed to identify the conserved domains of SPL family proteins. The gene structure of the GbSPL family was extracted from the *Ginkgo biloba* genome annotation file using TBtools-II (V2.210). The PlantCARE website tool (https://bioinformatics.psb.ugent.be/webtools/plantcare/html/, accessed on 1 April 2025) was used to predict cis-acting elements within 2000 bp upstream of the promoter sequences of the GbSPL genes. TBtools-II (V2.210) software was utilized for visualization.

### 4.5. Gene Duplication and Syntenic Analysis of the GbSPL Gene Family

Genome assemblies and annotation files for *Arabidopsis thaliana* and Poplar (*Populus alba* × *Populus tremula* var. *glandulosa* clone ‘84K’) genome data [52] were retrieved from Ensembl Plants (http://plants.ensembl.org/, accessed on 1 April 2025) and the National Center for Biotechnology Information (NCBI, https://www.ncbi.nlm.nih.gov/, accessed on 1 April 2025). Segmental duplication events between *Ginkgo biloba* and *Arabidopsis thaliana* or Poplar were identified and visualized through synteny analysis using TBtools-II (V2.210). The GbSPL gene replication events were analyzed using multiple collinear scanning toolkits (MCScanX, v1.0.0). The syntenic relationship between the GbSPL genes and SPL genes from *Arabidopsis thaliana* or Poplar was determined using Dual Synteny Plotter software (V2.210) in TBtools-II (V2.210).

### 4.6. RNA-Seq and Quantitative Real-Time PCR (qRT-PCR)-Based Expression Analysis of GbSPL Genes

The transcriptome data were downloaded from the National Center for Biotechnology Information database (NCBI, https://www.ncbi.nlm.nih.gov/, accessed on 1 April 2025) and then analyzed using the same method according to our previous study [39]. The accession number of different transcriptome data used in this study is listed in Appendix A. Heatmaps with the expression analysis of GbSPLs were generated by the TPM values with TBtools-II (V2.210). According to our previous study [38,39], a quantitative Real-Time PCR (qRT-PCR) assay was performed to confirm transcriptome sequencing results. Primer sequences used in this experiment are provided in Appendix A.

### 4.7. Subcellular Localization Analysis

Full-length coding sequences of *GbSPL4* and *GbSPL7* (excluding stop codons) were PCR-amplified from the cDNA of *Ginkgo biloba* leaves. The amplified products were directionally cloned into the pSAK277-GFP binary vector as C-terminal GFP fusions under the control of the CaMV 35S promoter. Constructs were heat-shocked into competent cells of *Agrobacterium tumefaciens* GV3101. The subcellular localization analysis followed our previously reported methods [38]. Primer sequences used in this experiment are provided in Appendix A.

## 5. Conclusions

The identification of 13 GbSPL genes categorized into six phylogenetic clusters indicates lineage-specific diversification, possibly reflecting adaptations to distinct environmental pressures faced by this ancient gymnosperm species. The prevalence of dispersed duplication events driving the expansion of the SPL family in *Ginkgo biloba* contrasts with the tandem duplication patterns typically observed in many angiosperms, suggesting different evolutionary mechanisms for gene family amplification in long-lived woody species. This divergence may contribute to Ginkgo’s well-known resilience to various biotic and abiotic stresses over extended periods. The discovery of miR156 targeting two GbSPL genes is consistent with conserved post-transcriptional regulatory mechanisms but raises questions regarding the functional redundancy or neofunctionalization of non-targeted GbSPLs. The nuclear localization of GbSPL2 and GbSPL11 reinforces their roles as transcription factors, likely involved in regulating downstream targets related to flavonoid biosynthesis and water stress response pathways. The contrasting expression patterns of GbSPLs under flavonoid induction suggest a complex regulatory network where specific SPL members function as either activators or repressors depending on developmental or environmental cues. This research not only enhances our understanding of the functional genomics of *Ginkgo biloba* but also lays the groundwork for utilizing SPL genes to boost stress tolerance and secondary metabolite production in economically significant gymnosperms. Overall, our genome-wide analysis of the SPL gene family lays the groundwork for further investigation into the mechanisms behind flavonoid biosynthesis and water stress in woody plants and gymnosperms.

## Figures and Tables

**Figure 1 ijms-26-04932-f001:**
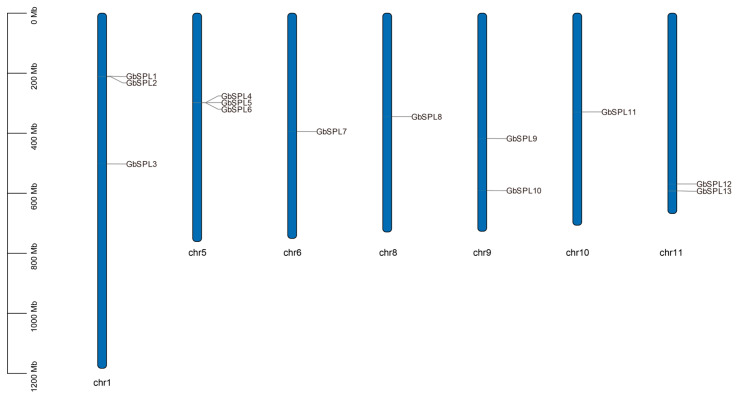
The chromosome distribution of *GbSPL* genes in the genome of *Ginkgo biloba*. The length of each chromosome was estimated in megabases (Mb).

**Figure 2 ijms-26-04932-f002:**
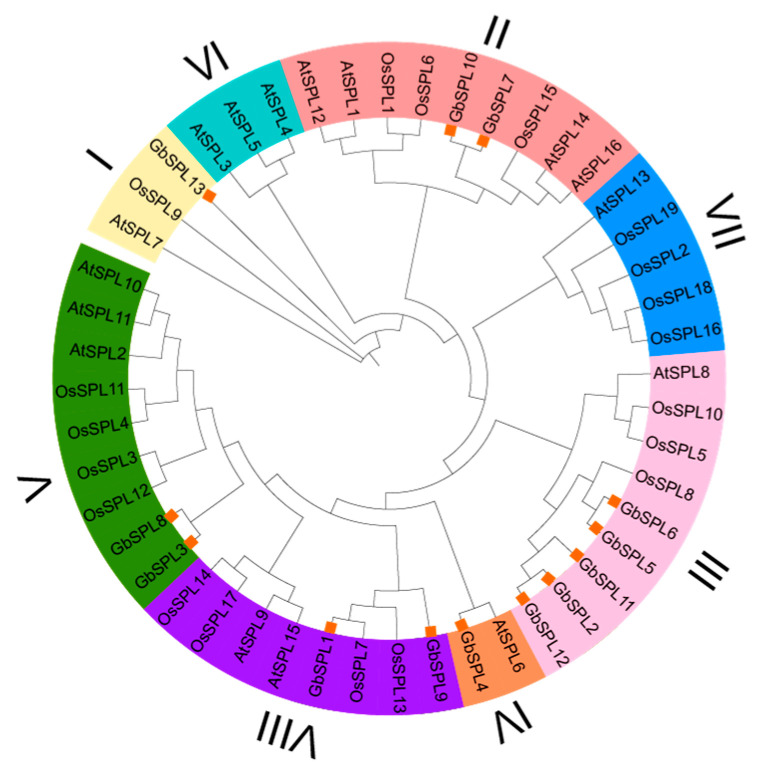
Phylogenetic analysis of GbSPL proteins with AtSPL proteins from Arabidopsis and OsSPL proteins from rice. The phylogenetic tree was constructed using IQ-Tree software, employing the Maximum Likelihood (ML) method, and bootstrap replications were set at 1000 times.

**Figure 3 ijms-26-04932-f003:**
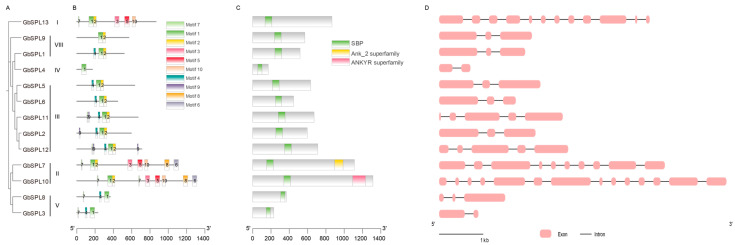
Phylogenetic trees, motifs, conserved domains, and gene structure analysis of GbSPL genes. (**A**) Phylogenetic trees of GbSPL proteins. (**B**) Motifs in the 13 GbSPL proteins, represented by ten colored boxes indicating different motifs. (**C**) Conserved domains present in the 13 GbSPL proteins. (**D**) Exon–intron structures of GbSPL genes.

**Figure 4 ijms-26-04932-f004:**
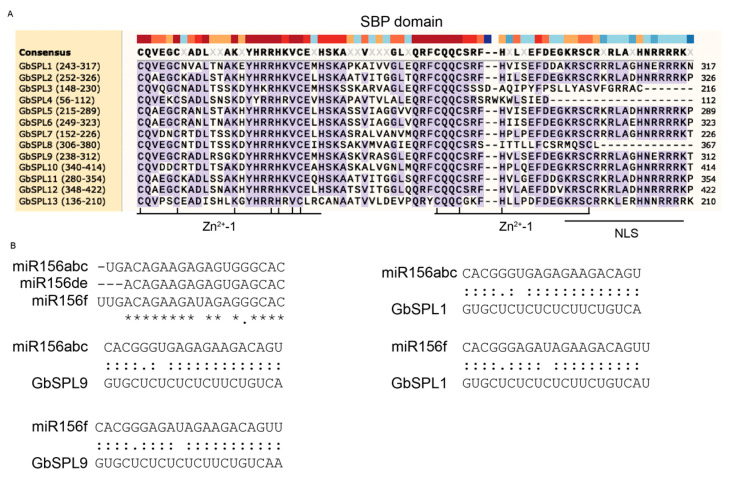
The sequence alignment of the SBP domain of GbSPL proteins and miR156 target gene binding sites. (**A**) The multiple sequence alignment of SBP domains. The two conserved zinc finger structures (C3H and C2HC) and NLS are indicated. The purple represents conserved bases. (**B**) The multiple sequence alignment of miR156 and its GbSPL target gene binding sites. The * represents conserved bases.

**Figure 5 ijms-26-04932-f005:**
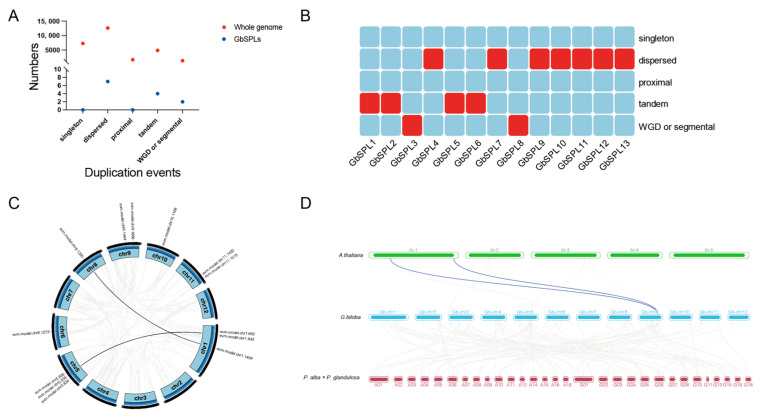
Evolutionary analysis of the GbSPL gene family. (**A**) The number of genes resulted from five duplication events, including singleton, dispersed, proximal, tandem, and WGD or segmental duplication. (**B**) The heatmap illustrates the duplication events of the GbSPL gene family. (**C**) The distribution and collinearity of the GbSPL gene family in the genome of *Ginkgo biloba*. The black line indicates the collinear gene pairs of GbSPL genes, while the gray line represents the collinear gene pairs in the *Ginkgo biloba* genome. (**D**) The distribution and collinearity of the GbSPL gene family in Arabidopsis, *Ginkgo biloba*, and Poplar. The blue line depicts the collinear gene pairs of GbSPL genes, whereas the gray line highlights the collinear gene pairs in the *Ginkgo biloba* genome with Arabidopsis and Poplar.

**Figure 6 ijms-26-04932-f006:**
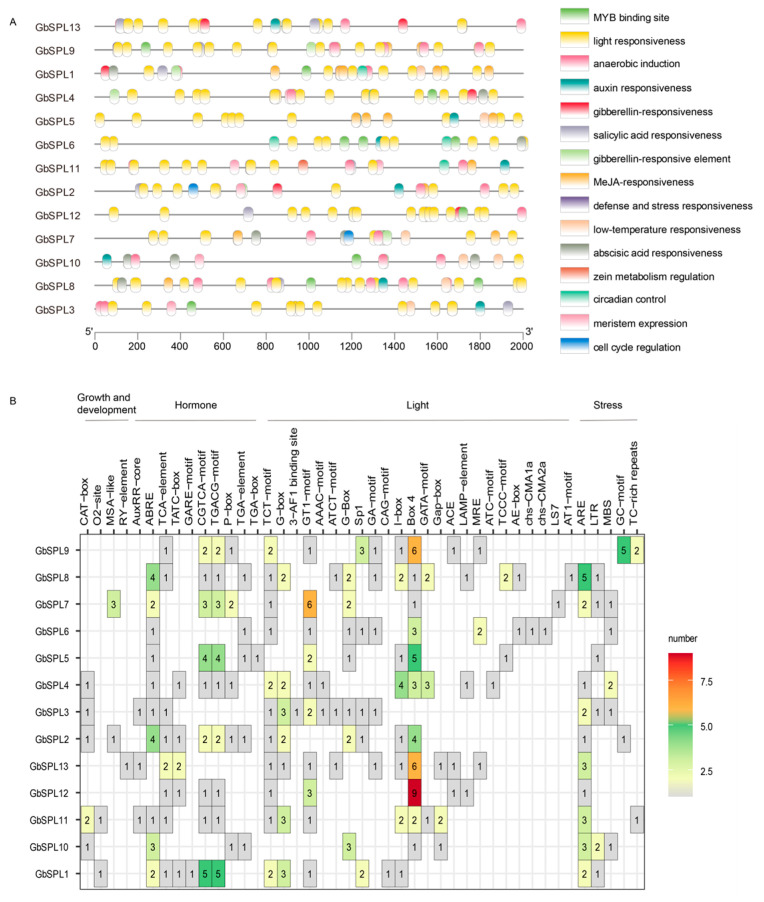
The cis-acting elements in the promoters of GbSPL genes. (**A**) The distribution of cis-acting elements in the promoters (upstream 2000 bp) of GbSPL genes. (**B**) The number of cis-acting elements among the four different types of response elements in the promoters of the GbSPL genes.

**Figure 7 ijms-26-04932-f007:**
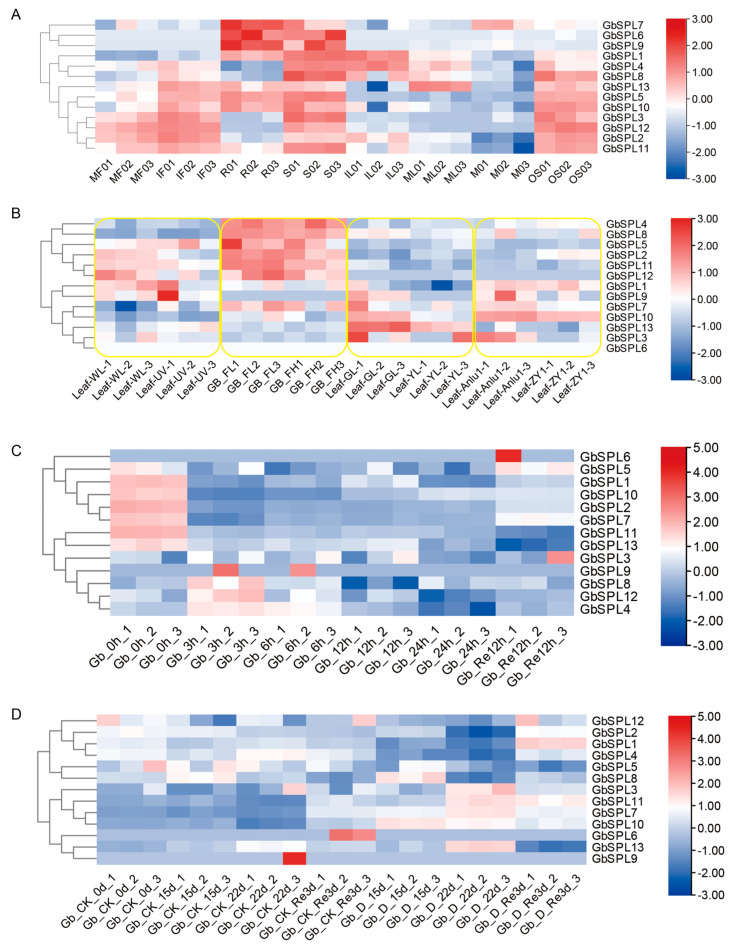
The heatmap illustrates the RNA-Seq-based expression patterns of 13 *GbSPL* genes. (**A**) Expression patterns of 13 *GbSPL* genes across eight tissues, including mature fruit (MF), immature fruit (IF), root (R), stem (S), immature leaf (IL), mature leaf (ML), microstrobilus (M), and ovulate strobilus (OS). (**B**) Expression patterns of the 13 *GbSPL* genes grouped by contrasting low and high flavonoid content. The yellow boxes represent different groups. (**C**,**D**) Expression patterns of 13 *GbSPL* genes under short-term (**C**) and long-term (**D**) water stress. Red indicates high-expression genes, while blue represents low-expression genes.

**Figure 8 ijms-26-04932-f008:**
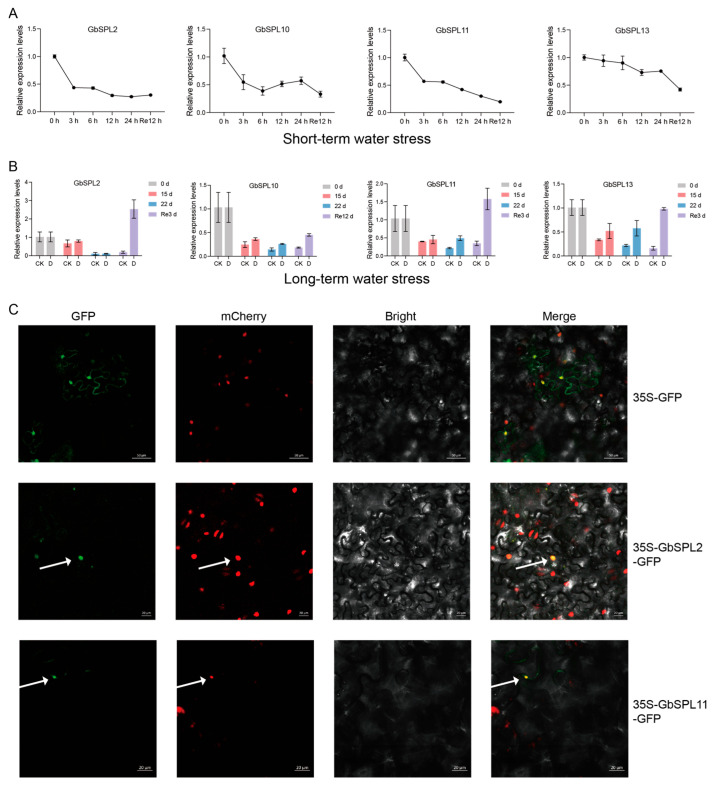
qRT-PCR analysis and subcellular localization of GbSPL genes. (**A**) qRT-PCR analysis of GbSPL genes under short-term water stress. All data are presented as means ± SD (n = 3). (**B**) qRT-PCR analysis of GbSPL genes under long-term water stress. (**C**) Subcellular localization of GbSPL2 and GbSPL11. mCherry, a nuclear marker. Arrows highlight fluorescent signals in the micrograph.

**Table 1 ijms-26-04932-t001:** Basic information of GbSPL gene family members in *Ginkgo biloba*.

Gene Name	Gene ID	Peptide (aa)	Molecular Weight MW (Da)	Theoretical pI	Grand Average of Hydropathicity (GRAVY)	Instability Index	Aliphatic Index	Subcellular Localization Predictions	miR156 Target
*GbSPL1*	evm.model.chr1.642	518	57,582.8	7.36	−0.776	58.94	58.90	nuclear	YES
*GbSPL2*	evm.model.chr1.643	596	64,192.75	7.10	−0.657	55.27	65.30	nuclear	NO
*GbSPL3*	evm.model.chr1.1404	230	25,387.53	8.98	−0.750	55.07	50.91	nuclear	NO
*GbSPL4*	evm.model.chr5.834	170	18,774.35	6.52	−0.514	51.55	79.24	nuclear	NO
*GbSPL5*	evm.model.chr5.835	633	70,406.88	7.61	−0.765	65.35	60.39	nuclear	NO
*GbSPL6*	evm.model.chr5.836	447	49,707.51	8.55	−0.654	65.54	64.41	nuclear	NO
*GbSPL7*	evm.model.chr6.1273	1112	123,372.1	8.56	−0.499	55.69	75.50	nuclear	NO
*GbSPL8*	evm.model.chr8.1250	367	40,125.93	7.52	−0.622	37.22	58.96	nuclear	NO
*GbSPL9*	evm.model.chr9.1444	569	63,284.01	9.27	−0.707	53.76	66.40	nuclear	YES
*GbSPL10*	evm.model.chr9.1866	1314	145,251.18	8.69	−0.414	58.05	78.76	chloroplast	NO
*GbSPL11*	evm.model.chr10.1109	670	72,574.69	8.92	−0.741	59.74	60.70	nuclear	NO
*GbSPL12*	evm.model.chr11.1430	711	77,686.16	8.58	−0.693	56.53	58.30	nuclear	NO
*GbSPL13*	evm.model.chr11.1518	866	97,933.52	5.57	−0.523	50.55	76.74	nuclear	NO

## Data Availability

The public transcriptome data used in this research can be accessed in the inserted article.

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
