# Peer review of "Genome-Wide Identification of Ginkgo biloba SPL Gene Family and Expression Analysis in Flavonoid Biosynthesis and Water Stress"

_ijms, 2025, doi:10.3390/ijms26104932_

Round 1
Reviewer 1 Report
Comments and Suggestions for Authors
Abstracts
In 23-25 lines, “GbSPL genes possess motifs associated with light, hormones, and stress,” what hat is the direct relationship between this sentence and the flavonoids synthesis?
Introduction
For introduction part, the functional explanation of SPL genes is not sufficient. it is necessary to mention which crops they have been identified in, and describe approximately how many members they have, their classification, and what functions they perform.
In last paragraph of introduction, it should be simply summarized the research content of this article, whether is too verbose and complex in this manuscript.
Results
For the Table 1, “subcellular localization” should be “subcellular localization predictions”.
For Figure 1, the “chr1~chr11”shoule be in end.
Authors use two plant species to complete a phylogenic tree, why not choose more plant species for the construction phylogenic tree?
Materials and methods
Please provide the basis for naming of the SPL member family in Ginkgo biloba.
Comments on the Quality of English Languageno
Author Response
Comments 1: Abstracts
In 23-25 lines, “GbSPL genes possess motifs associated with light, hormones, and stress,” what hat is the direct relationship between this sentence and the flavonoids synthesis?
Response 1: Thanks for pointing this out. The motifs in the promoter of GbSPL genes are associated with light, hormones, and stress responses, which suggests that GbSPL genes might be involved in pathways that respond to those signals. Flavonoid biosynthesis is often regulated by environmental factors like light and stress, as well as hormonal signals [1,2]. For example, light can induce flavonoid production through photoreceptors, and stress signals like pathogens or UV light can trigger flavonoid synthesis as a protective measure. Hormones like jasmonic acid or ethylene might also play roles in regulating these pathways. Thus, the direct relationship is that GbSPL genes, being regulated by light, hormones, and stress through their motifs, act as regulatory switches that activate or enhance flavonoid biosynthesis pathways in response to these environmental and internal cues. This would help the plant adapt by producing flavonoids when needed, such as under stress or specific light conditions.
- Shen, N.; Wang, T.; Gan, Q.; Liu, S.; Wang, L.; Jin, B. Plant flavonoids: Classification, distribution, biosynthesis, and antioxidant activity. Food Chem 2022, 383, 132531, doi:10.1016/j.foodchem.2022.132531.
- Liu, W.; Feng, Y.; Yu, S.; Fan, Z.; Li, X.; Li, J.; Yin, H. The Flavonoid Biosynthesis Network in Plants. Int J Mol Sci 2021, 22, doi:10.3390/ijms222312824.
Comments 2: Introduction
For introduction part, the functional explanation of SPL genes is not sufficient. it is necessary to mention which crops they have been identified in, and describe approximately how many members they have, their classification, and what functions they perform.
Response 2: Thanks for your nice suggestion. We have described “which crops they have been identified in, and describe approximately how many members they have” in lines 48-58. In addition, we have added more description on their classification and what functions they perform in the revised manuscript, please see lines 59-60 and 87-97.
Comments 3: In last paragraph of introduction, it should be simply summarized the research content of this article, whether is too verbose and complex in this manuscript.
Response 3: Thanks for your nice suggestion. We have shortened the summarized content in the last paragraph of the introduction in the revised manuscript, please see lines 112-115.
Results
Comments 4: For the Table 1, “subcellular localization” should be “subcellular localization predictions”.
Response 4: Thanks for your nice suggestion. We have changed the “subcellular localization” to “subcellular localization predictions” in the revised Table 1.
Comments 5: For Figure 1, the “chr1~chr11”shoule be in end.
Response 5: Thanks for your nice suggestion. We have moved the “chr1~chr11” to the end in the revised Figure 1.
Comments 6: Authors use two plant species to complete a phylogenic tree, why not choose more plant species for the construction phylogenic tree?
Response 6: Thank you for the constructive suggestion. We have added one more plant species (rice) for the construction phylogenetic tree in the revised manuscript, please see the revised Figure 2 (line 161). A dicotyledonous model plant, Arabidopsis thaliana, a monocotyledonous model plant, rice, and a gymnosperm, Ginkgo biloba, were selected for phylogenetic tree analysis. The phylogenetic tree results obtained from these plants are representative.
Comments 7: Materials and methods
Please provide the basis for naming of the SPL member family in Ginkgo biloba.
Response 7: Thanks for your nice suggestion. We have added more details on the basis for naming the SPL member family in Ginkgo biloba in the revised manuscript, please see lines 460-462.
Reviewer 2 Report
Comments and Suggestions for Authors
Dear Authors,
I have reviewed the manuscript and have the following comments:
The topic of this manuscript is the genome-wide identification and expression analysis of the SPL gene family in an important species, Ginkgo biloba, which has been studied in flavonoid biosynthesis. The authors found that analysis of gene structure, conserved domains, conserved motifs, and miR156 target prediction indicate that GbSPLs are highly conserved through evolutionary timelines. Furthermore, synthesis analysis highlighted that dispersed duplication events expanded the SPL gene family in Ginkgo biloba.
The manuscript is significant because Ginkgo biloba is an important species in ornamental horticulture and food production, and its research is yielding valuable results. Its genomics research is also very important for further breeding work.
The manuscript is novel in that it offers the first systematic characterization of the Ginkgo biloba SPL gene family, providing a valuable basis for understanding their evolutionary background and functional role in flavonoid biosynthesis and response to water stress.
The manuscript is in English, figures and tables are correct.
However, I suggest the following modifications:
Introduction: I request a longer paragraph on the Ginkgo biloba species because it is only mentioned as part of the hypothesis, I think it deserves much more than that and there are valuable results about it.
Conclusions: I suggest rewriting the chapter, as a summary is not needed here, but rather the implications of the manuscript's results should be presented.
Author Response
Comments 1: Dear Authors,
I have reviewed the manuscript and have the following comments:
The topic of this manuscript is the genome-wide identification and expression analysis of the SPL gene family in an important species, Ginkgo biloba, which has been studied in flavonoid biosynthesis. The authors found that analysis of gene structure, conserved domains, conserved motifs, and miR156 target prediction indicate that GbSPLs are highly conserved through evolutionary timelines. Furthermore, synthesis analysis highlighted that dispersed duplication events expanded the SPL gene family in Ginkgo biloba.
The manuscript is significant because Ginkgo biloba is an important species in ornamental horticulture and food production, and its research is yielding valuable results. Its genomics research is also very important for further breeding work.
The manuscript is novel in that it offers the first systematic characterization of the Ginkgo biloba SPL gene family, providing a valuable basis for understanding their evolutionary background and functional role in flavonoid biosynthesis and response to water stress.
The manuscript is in English, figures and tables are correct.
Response 1: We greatly appreciate the positive comments and valuable suggestions from the reviewer.
Comments 2: However, I suggest the following modifications:
Introduction: I request a longer paragraph on the Ginkgo biloba species because it is only mentioned as part of the hypothesis, I think it deserves much more than that and there are valuable results about it.
Response 2: We greatly appreciate your constructive suggestions. We have added more content on the Ginkgo biloba species in the revised manuscript. Please see lines 103-110.
Comments 3: Conclusions: I suggest rewriting the chapter, as a summary is not needed here, but rather the implications of the manuscript's results should be presented.
Response 3: Thanks for your nice suggestion. We have rewritten the Conclusions and added more content on the implications of the manuscript's results in the revised manuscript. Please see lines 516-534.